# Sleep Duration on Workdays Is Correlated with Subjective Workload and Subjective Impact of High Workload on Sleep in Young Healthy Adults

**DOI:** 10.3390/brainsci13050818

**Published:** 2023-05-18

**Authors:** Charlotte von Gall, Thomas Muth, Peter Angerer

**Affiliations:** 1Institute of Anatomy II, Medical Faculty, Heinrich Heine University, 40225 Düsseldorf, Germany; 2Institute for Occupational, Social and Environmental Medicine, Medical Faculty, Heinrich Heine University, 40225 Düsseldorf, Germany

**Keywords:** actigraphy, wearable, sleep quality, psychosocial stress, Fitbit

## Abstract

Psychosocial stress is widespread worldwide and particularly affects young adults. There is a close and bidirectional relationship between sleep quality and mental health. Sleep duration, which is an important feature of sleep quality, shows both intra-individual variations and inter-individual differences. Internal clocks control individual sleep timing, which, in turn, defines the chronotype. On workdays, however, the end and duration of sleep are largely limited by external factors, such as alarm clocks, especially in later chronotypes. The aim of this study is to investigate whether there is a relationship between sleep timing and duration on workdays and measures for psychosocial stress, such as anxiety and depression; subjective workload; and the subjective impact of a high workload on sleep. We used a combination of Fitbit wearable actigraphy data and a questionnaire survey of young, healthy medical students and calculated correlations between the respective variables. We found that a shorter sleep duration on workdays is associated with a higher subjective workload and a higher subjective impact of a high workload on sleep, which, in turn, are associated with higher measures of anxiety and depression. Our study contributes to understanding the importance of sleep timing/duration and their regularity on weekdays for subjectively perceived psychosocial stress.

## 1. Introduction

Mood-related disorders such as anxiety and depression are highly prevalent globally and disproportionately affect young adults [1]. Thus, the mental health of young adults and students in higher education, in particular, remains a matter of concern [2,3,4]. Psychosocial factors such as stress, depression, and anxiety have already been well examined in medical students [5,6]. Study-related stressors experienced by medical students include high workloads and tight time schedules [7,8]. Subjective workload has a strong impact on mental health in general [9,10], as well as on motivation and academic performance [11]. Moreover, psychological stress is a risk factor for various psychosomatic diseases, such as cardiovascular diseases [12], and gut disorders [13], and is associated with muscle parafunctional activity [14].

Sleep plays a crucial role in health in general and mental health in particular [15]. Sleep problems, which are highly prevalent in modern society, are associated with multiple adverse consequences [16]. Sleep duration, which is an important attribute of sleep quality [17], shows substantial intra- and inter-individual variations [15]. Large-scale cross-sectional studies show that sleep duration is associated with general health, as well as the physical and mental quality of life [18]. Although the relationship between sleep and mood-related factors is bidirectional, there is evidence that the effect of sleep quality on mood is greater than vice versa [19]. Importantly, daily rhythms in sleep and wake are controlled by a complex interplay of different oscillators, including exogenous oscillators, such as light–dark and social cycles, and endogenous oscillators, such as circadian and homeostatic oscillators [20,21]. According to the endogenous oscillators, individuals differ in their intrinsically determined preferred sleep timing, defining the chronotype [22]. However, in modern society, sleep duration (SD) and sleep timing on workdays are mainly determined by external factors, such as the morning alarm clock. Thus, particularly in later chronotypes, early work schedules result in short sleep duration on workdays. This could be one reason why sleep duration has decreased in many countries over the past 100 years [23] and also why sleep quality is lower on workdays than on free days [24]. There is increasing evidence that sleep duration is a critical driver of major depression [25], and later chronotypes demonstrate higher levels of depression and sleep disturbances [26,27]. In addition, intra-individual variability in sleep duration as a result of shift work substantially impacts mood and depression [28]. In outpatient studies, sleep quality and timing are typically evaluated by questionnaires, which do not distinguish between workdays and free days. However, recent research shows that sleep quality differs between weekdays and workdays [24], so it is important to make this distinction. In addition, a questionnaire does not allow longitudinal data collection, which is a prerequisite for the analysis of regularity. Wearable actigraphy offers a great opportunity to longitudinally study the relationships between sleep timing, its inter- and intra-individual variation, and mental health under real-life conditions [25,26]. Using a combination of Fitbit wearable actigraphy data and a questionnaire survey of young, healthy medical students, we were recently able to show that the discrepancy between sleep duration on workdays and free days is associated with higher fragmentation of REM sleep, which, in turn, is associated with higher subjective tiredness upon waking [29].This indicates that limiting sleep duration by starting work early has a negative impact on objective sleep architecture and subjective sleep quality. Here, we examine other variables from the same dataset. Our approach, using Fitbit wearable actigraphy in combination with questionnaires, allows for the analysis of correlations of objective measures of sleep timing/duration and their regularity on workdays and free days with subjective measures of psychosocial stress, such as anxiety and depression; subjective workload; and the subjective impact of a high workload on sleep. With this study, we contribute to understanding the importance of sleep timing/duration and their regularity on weekdays for subjectively perceived psychosocial stress.

## 2. Materials and Methods

### 2.1. Ethics and Participants

The questionnaire surveys and the assessments of Fitbit sleep data in healthy subjects are in agreement with the ethical requirements of the Declaration of Helsinki and were approved by the Research Ethics Committee of the Medical Faculty (ChronoSleep study consent number: 2019-3786). Exclusion criteria were (i) age below 18 years, (ii) shift work, (iii) work on weekends, (iv) chronic diseases including sleep disorders, and (v) chronic medication including sleep medication. All subjects provided informed consent after receiving a complete description of the study. The data used for this study came from the same dataset as from a previous study [29]. For this dataset, medical students in their first year were invited to participate between 25 June 2021 and 19 May 2022. Importantly, during this period, most of the courses did not start before 11 am because, due to the COVID-19 pandemic, all lectures were provided as screencasts and could be attended by the students according to their own time schedule.

### 2.2. Study Design

Volunteers received a pseudonym and were equipped with a Fitbit Inspire multisensory (motion and heart rate) sleep-tracking device (San Francisco, CA, USA). They were asked to wear it for 90 days, i.e., 64 workdays and 26 free days, as continuously as possible, especially at night. After the data collection period of 90 days, participants filled out the online questionnaire and authorized sleep data transfer from Fitbit to our study server. All questions in the survey were mandatory and could only be answered once. Data from the questionnaire and Fitbit were assigned by a pseudonym.

### 2.3. Measures, Data Processing, and Statistics

The online questionnaire included items on demographics; workload (WL) (“How high would you rate your workload in general?”); the impact of workload on sleep (WLI) (“Does the heavy workload have a negative effect on your sleep quality?”); cohabitation (cohab) (“Do you live alone or together with family, shared apartment, friends, spouse”); alarm clock use on workdays and free days; and general health, including the 4-item depression and anxiety screen (PHQ-4) and others [29]. The scores and distribution of the respective variables assessed by the questionnaire are shown in Table 1.

A customized software application was used to transfer and pre-process Fitbit sleep data [29]. The following parameters were separately calculated based on Fitbit sleep data for workdays (weekdays) and free days (weekends): (i) sleep duration (SD) in a decimal format of hours, (ii) intra-individual variation in sleep duration as standard deviation (SD_std), and (iii) start and end time of sleep in a decimal format of clock time. The decimal format of hours/clock time was calculated by dividing the minutes by 60.

Statistical analyses were performed using Prism 7.01 (Graph Pad, Boston, MA, USA). Data were expressed as a percentage of the total sample or mean +/− standard deviation. A confidence interval of 95% was used consistently. The Gaussian distribution was analyzed using the D’Agostino–Pearson normality test. Differences between workdays and free days were analyzed using paired *t*-tests. If variables were normally distributed, correlations were analyzed using Pearson’s test, and linear regression was performed. Correlations between variables, of which at least one was not normally distributed, were calculated using the Spearman test. *p*-values < 0.05 were considered statistically significant.

## 3. Results

### 3.1. Sample Characteristics

The sample characteristics are described in detail in [29]. Briefly, the sample consisted of 59 participants, of which 43 were women (73%). The high proportion of women reflects the gender distribution among all medical students of the year. The participants were 21.18 +/− 2.36 years old. A total of 82.4 (+/− 9.7) days of Fitbit sleep data were recorded. According to the questionnaire, all participants used an alarm clock on workdays, while only 15 participants (25%) used an alarm clock on free days.

### 3.2. Variables Assessed by Questionnaire

The definition and distribution of variables assessed by the questionnaire are summarized in Table 1. Most of the participants lived in the community (64%) and reported a high (52%) or very high (5%) workload (WL). A little less than half reported that a high WL had an impact on their sleep (WLI) (41%). A small proportion of participants showed moderate (10%) or severe (7%) symptoms of anxiety and depression. Only WL passed the normality test.

Subjective workload (WL) is positively correlated with the subjective impact of workload on sleep (WLI) (r = 0.73, *p* < 0.0001). Thus, a higher subjective workload is associated with a higher subjective impact of workload on sleep.

Depression and anxiety (PHQ4) are positively correlated with both WL (r = 0.3, *p* = 0.02) and WLI (r = 0.34, *p* = 0.009) and negatively correlated with living in the community (cohab) (r = −0.27, *p* = 0.04). Thus, a higher score for depression and anxiety is associated with a higher subjective workload and a higher subjective impact of a high workload on sleep.

### 3.3. Objective Measures for Sleep Timing on Workdays and Free Days Longitudinally Assessed by Wearable Actigraphy

All of the following sleep duration and sleep timing variables passed the normality test.

Sleep duration on workdays (SD on WDs, 7.89 +/− 0.67) is significantly shorter (*t* = 2.57, df = 58, *p* = 0.012) than sleep duration on free days (SD on FDs, 8.09 +/− 0.72). SD on WD and SD on FD are positively correlated (r = 0.66, *p* < 0.0001). On WD, three participants (5%) had a sleep duration shorter than 7 h, while on FD, the sleep duration was longer than 7 h for all participants.

Intra-individual variation (_std) in sleep duration is significantly lower (*t* = 4.88, df = 58, *p* < 0.0001) on workdays (SD_std on WDs, 1.25 +/− 0.35) than on free days (SD_std on FDs, 1.46 +/− 0.40). SD_std on WD and SD_std on FD are positively correlated (r = 0.65, *p* < 0.0001).

There is no significant correlation between sleep duration and intra-individual variation in sleep duration on workdays (Figure 1a) or free days (Figure 1b).

Sleep onset is significantly earlier (*t* = 11.99, df = 58, *p* < 0.0001) on workdays (00:04 +/− 0.89) than on free days (00:92 +/− 0.97). Additionally, sleep offset is significantly earlier (*t* = 12.88, df = 58, *p* < 0.0001) on workdays (08:01 +/− 0.72) than on free days (09:02 +/− 0.87).

On workdays, SD is negatively correlated with the time of sleep onset (r = −0.58, *p* < 0.0001; Figure 2a). Thus, later sleep onset is associated with shorter sleep duration on workdays. On free days, SD is negatively correlated with the time of sleep onset (r = −0.40, *p* = 0.002) and positively correlated with the time of sleep offset (r = 0.34, *p* = 0.008; Figure 2b). Thus, longer sleep duration is associated with both earlier sleep onset and later sleep offset on free days.

Intra-individual variation in SD (SD_std) on workdays is positively correlated with the time of sleep onset (r = 0.26, *p* = 0.045) and offset (r = 0.34, *p* = 0.008) on free days (Figure 3a). Thus, later sleep timing on free days is associated with higher intra-individual variation in sleep duration on workdays. Intra-individual variation in SD (SD_std) on free days is not significantly correlated with the time of sleep onset or offset (Figure 3b).

### 3.4. Correlations between Sleep Duration and Variables Assessed by Questionnaire

Subjective workload (WL) is negatively correlated with SD on WD (r = −0.28, *p* = 0.03) but not significantly correlated with SD on FD (r = −0.19, *p* = 0.13; Figure 4a).

The subjective impact of workload on sleep is negatively correlated with SD on WD (r = −0.29, *p* = 0.03) and SD on FD (r = −0.29, *p* = 0.03; Figure 4b). Thus, a shorter sleep duration on workdays is associated with both a higher subjective workload and a higher subjective impact of a high workload on sleep, while a shorter sleep duration on free days is associated with a higher subjective impact of a high workload on sleep.

PHQ4 is negatively correlated with SD_std on FD (r = −0.38, *p* = 0.003) but not significantly correlated with SD_std on WD (r = −0.19, *p* = 0.14; Figure 5a).

Living in the community (cohab) is positively correlated with SD_std on WD (r = 0.29, *p* = 0.03) but not significantly correlated with SD_std on FD (r = 0.18, *p* = 0.17; Figure 5b).

In summary, sleep duration on workdays, which is mainly determined by sleep onset and restricted by sleep offset, is interconnected with the subjective measures of psychosocial stress (Figure 6).

## 4. Discussion

Consistent with our previous study on sleep timing and sleep composition in this cohort [29] and expectations from the literature [30], sleep duration is shorter and sleep time is significantly earlier on workdays than on free days. On free days, sleep duration is correlated with both the time of sleep onset and offset. On workdays, sleep duration is only correlated with the time of sleep onset. This is consistent with sleep duration on workdays being constrained by external factors, such as alarm clocks in the morning. This is, nevertheless, surprising because the participants in our study were free to choose the start of their working hours, as most morning lectures were provided as screencasts due to the COVID-19 pandemic. This indicates a high degree of discipline and is probably due to the fact that getting up late is fraught with prejudice in our society. The lower intra-individual variability in sleep duration on workdays compared to free days might also be a consequence of the constrained time of sleep offset on workdays. The positive correlation of sleep time on free days with intra-individual variability in sleep duration on workdays suggests that later chronotypes show greater variability in their sleep duration on workdays.

The results of PHQ4 screening for depression and anxiety in our cohort were comparable to a larger cohort study also conducted during the COVID-19 pandemic at another major German medical school [5]. There seems to be a sustained association between COVID-19-related stress [31] and negative mental health outcomes [32,33] but also with positive mental health outcomes [33]. Notably, a lower PHQ4 score is associated with living in the community. Living alone may have a particularly negative reinforcing effect in our cohort because during the COVID-19 pandemic, social interactions were largely reduced to flatmates, and loneliness and stress responses, as well as symptoms of depression/anxiety, were increased in adults living alone [34] or with fewer resources [35]. Living in the community and a lower PHQ4 score are associated with higher intra-individual variability in sleep duration on workdays and free days, respectively. This suggests that in our study, irregular sleep duration is more a consequence of social interactions due to living in the community and evening activities on weekends, despite COVID-19-related social confinement. Importantly, in our study, which *excludes shift work*, intra-individual variation in sleep duration has no negative effects on psychosocial health. In contrast, for shift work, intra-individual variability in sleep duration is correlated with depression and anxiety [25], which is consistent with the aversive effect of shift work on mental health [36,37] and health in general [38]. In this context, we would like to emphasize that we examined a relatively homogeneous cohort of young, healthy adults that did not show any major deviations. In contrast to other studies [15,26,39,40], we did not find a direct correlation between sleep duration and depression scores. This may be because the sample size in our study is too small to observe this effect or because, in contrast to recording sleep duration using questionnaires, we measured it objectively, longitudinally, and separately for workdays and free days. We were able to establish a positive correlation between PHQ4 and subjective workload and the subjective impact of a high workload on sleep, both of which are negatively correlated with sleep duration on workdays. In addition, the subjective impact of a high workload on sleep is negatively correlated with sleep duration on free days. Thus, the perception of a high workload and its subjective effect on sleep, both of which have a negative impact on depression and anxiety, are associated with shorter sleep duration. This is consistent with the association of short sleep with a high mental workload [41] and the association of psychosocial stress with poor sleep quality [42,43]. The reciprocal relationship between stress, which activates the hypothalamic–pituitary–adrenal axis and sympathetic nervous system, and insomnia is well known [44]. However, it is still surprising that even for normal sleepers, there is a relationship between a slight reduction in the duration of sleep on workdays compared to days off and subjective stress levels. In addition, we would like to point out that although all participants have a comparable objective workload, the subjective workload should not be underestimated, as it has a strong influence not only on mental health [8,9] but also on academic motivation and performance [10]. Thus, our study also provides a new approach to elucidate the complex interaction between sleep timing and academic performance [45].

## 5. Conclusions

Our study suggests that a longer sleep duration on workdays is associated with lower levels of psychosocial stress and, hence, better mental health. Since the duration of sleep on workdays is largely determined by the endogenous chronotype, modern concepts of education and work should allow for flexible and individual working hours.

## Figures and Tables

**Figure 1 brainsci-13-00818-f001:**
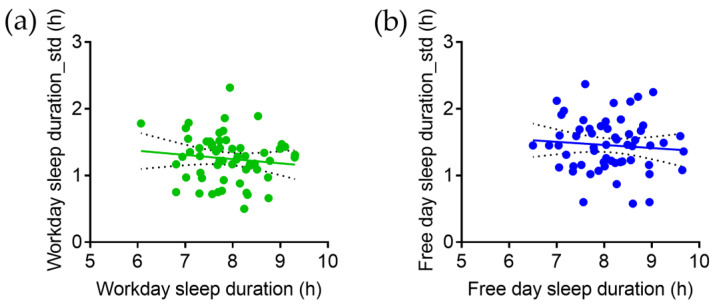
Correlation between sleep duration and intra-individual variation (_std) on (**a**) workdays and (**b**) free days.

**Figure 2 brainsci-13-00818-f002:**
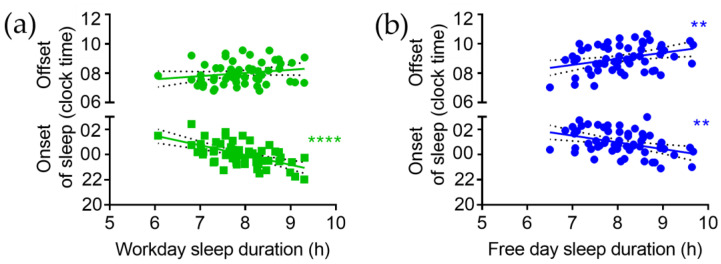
Correlation of sleep duration with sleep onset and sleep offset on (**a**) workdays and (**b**) free days. ** *p* < 0.01, **** *p* < 0.0001; slope of linear regression is significantly different from zero.

**Figure 3 brainsci-13-00818-f003:**
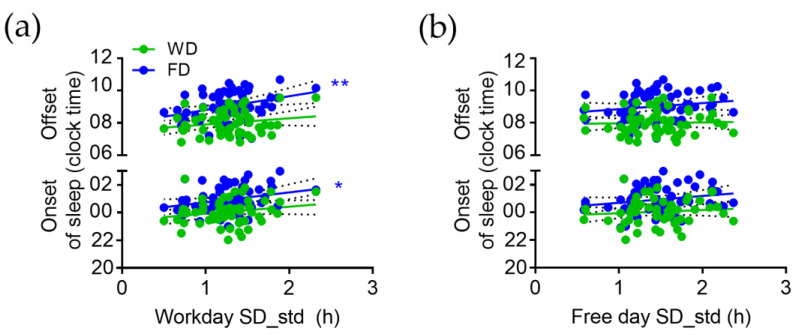
Correlation of intra-individual variation in sleep duration (SD_std) on (**a**) workdays and (**b**) free days with sleep onset and sleep offset. * *p* < 0.05, ** *p* < 0.01; slope of linear regression is significantly different from zero.

**Figure 4 brainsci-13-00818-f004:**
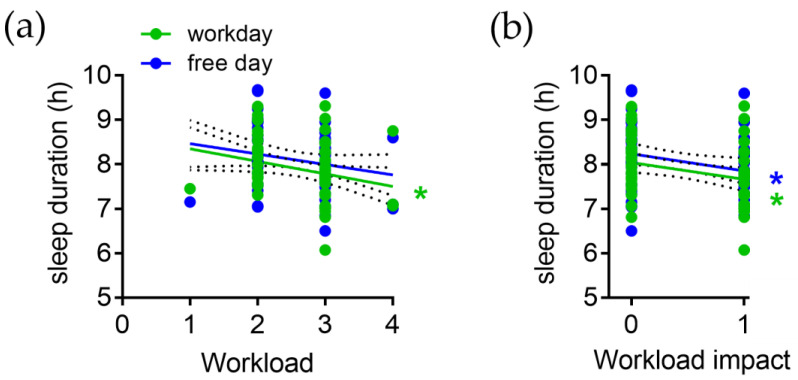
Correlation of sleep duration on workdays (green) and free days (blue) with (**a**) subjective workload and (**b**) subjective impact of high workload on sleep. * *p* < 0.05, slope of linear regression is significantly different from zero.

**Figure 5 brainsci-13-00818-f005:**
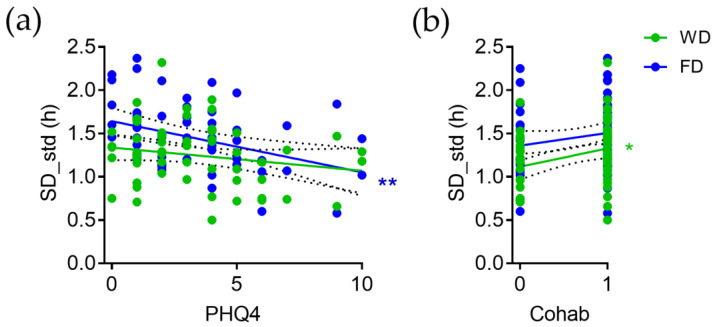
Correlation of intra-individual variation in sleep duration (SD_std) on workdays ((WDs) green) and free days ((FDs) blue) with (**a**) symptoms of depression and anxiety (PHQ4) and (**b**) living in the community (cohab). * *p* < 0.05, ** *p* < 0.01; slope of linear regression is significantly different from zero.

**Figure 6 brainsci-13-00818-f006:**
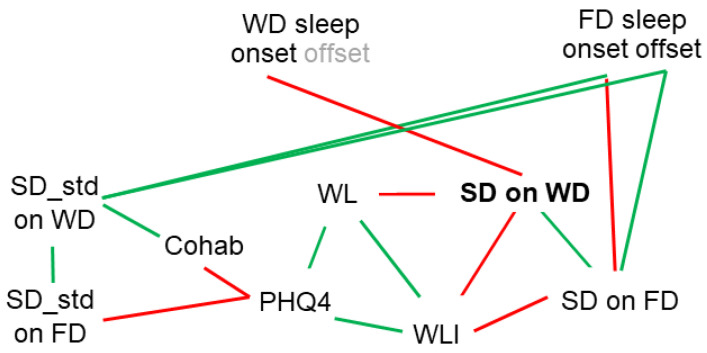
Graphic summary of relationships between sleep timing assessed by Fitbit wearable actigraphy and subjective measures for psychosocial stress assessed by questionnaires. Negative correlations are indicated in red, positive correlations are indicated in green. Time of sleep *onset* on workdays (WDs) and free days (FDs) is negatively correlated with sleep duration (SD) on WD and FD, respectively. Time of sleep *offset* on FD is positively correlated with SD on FD. SD on WD is positively correlated with SD on FD and negatively correlated with subjective workload (WL) and subjective impact of high workload on sleep (WLI). WL and WLI are positively correlated, and both are positively correlated with symptoms of anxiety and depression (PHQ4). PHQ4 is negatively correlated with living in the community (cohab). Intra-individual variation in sleep duration (SD_std) on workdays is positively correlated with sleep onset and offset on FD, SD_std on free days, and cohab. SD_std on FD is negatively correlated with PHQ4.

**Table 1 brainsci-13-00818-t001:** Scoring and distribution of variables assessed by questionnaire. Standard deviation (std) shows the inter-individual variation.

Variable	Score	Mean +/− std	Total (%)
**Cohabitation**		0.64 +/− 0.48	
Living alone	0		21 (36%)
Living in the community	1		40 (64%)
Workload		2.6 +/− 0.62	
Low	1		1 (2%)
Medium	2		24 (41%
High	3		31 (52%)
Very high	4		3 (5%)
Workload impact on sleep		0.41 +/− 0.49	
No	0		35 (59%)
Yes	1		24 (41%)
Anxiety and depression		3.18 +/− 2.56	
Normal	0–2		29 (49%)
Mild	3–5		20 (34%)
Moderate	6–8		6 (10%)
Severe	9–12		4 (7%)

## Data Availability

Data are unavailable due to privacy or ethical restrictions.

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
