# Peer review of "Sleep Duration on Workdays Is Correlated with Subjective Workload and Subjective Impact of High Workload on Sleep in Young Healthy Adults"

_brainsci, 2023, doi:10.3390/brainsci13050818_

Round 1
Reviewer 1 Report
The manuscript entitled " Sleep duration on workdays is correlated with subjective workload and subjective impact of high workload on sleep in young healthy adults" has been investigated in detail. The topic addressed in the manuscript is potentially interesting and the manuscript contains some practical meanings, however, there are some issues which should be addressed by the authors:
1) In the first place, I would encourage the authors to extend the abstract more with the key results. As it is, the abstract is a little thin and does not quite convey the interesting results that follow in the main paper. The "Abstract" section can be made much more impressive by highlighting your contributions. The contribution of the study should be explained simply and clearly.
2) The readability and presentation of the study should be further improved. The paper suffers from language problems.
3) The "Introduction" section needs a major revision in terms of providing more accurate and informative literature review and the pros and cons of the available approaches and how the proposed method is different comparatively. Also, the motivation and contribution should be stated more clearly.
4) The importance of the design carried out in this manuscript can be explained better than other important studies published in this field. I recommend the authors to review other recently developed works.
5) What makes the proposed method suitable for this unique task? What new development to the proposed method have the authors added (compared to the existing approaches)? These points should be clarified.
6) "Discussion" section should be added in a more highlighting, argumentative way. The author should analysis the reason why the tested results is achieved.
2) The readability and presentation of the study should be further improved. The paper suffers from language problems.
Author Response
Thank you for your revision, which greatly improved the quality of the manuscript. The suggested changes are highlighted in the revised manuscript.
1) In the first place, I would encourage the authors to extend the abstract more with the key results. As it is, the abstract is a little thin and does not quite convey the interesting results that follow in the main paper. The "Abstract" section can be made much more impressive by highlighting your contributions. The contribution of the study should be explained simply and clearly.
Response: Unfortunately, the abstract only allows 200 words. The key results are summarized in the in the penultimate sentence. The contribution of the study is now explained in the (edited) last sentence.
2) The readability and presentation of the study should be further improved. The paper suffers from language problems.
Response: Language editing will be performed
3) The "Introduction" section needs a major revision in terms of providing more accurate and informative literature review and the pros and cons of the available approaches and how the proposed method is different comparatively. Also, the motivation and contribution should be stated more clearly.
Response: We added additional literature and the pros and cons of the relevant approaches.
4) The importance of the design carried out in this manuscript can be explained better than other important studies published in this field. I recommend the authors to review other recently developed works.
Response: Thank you. We reviewed other recently developed works.
5) What makes the proposed method suitable for this unique task? What new development to the proposed method have the authors added (compared to the existing approaches)? These points should be clarified.
Response: We clarified these points.
6) "Discussion" section should be added in a more highlighting, argumentative way. The author should analysis the reason why the tested results is achieved.
Response: We added a reason why the tested results are achieved.
Reviewer 2 Report
The work in my opinion interesting, congratulates the idea of the authors. I have the following comments
Abstract
L18 – ‘’We used a combination of Fitbit wearable actigraphy data and ‘’ - I suggest adding how much actigraphy was carried for.
L22 -23 - The last sentence of the abstract suggests deleting. It does not contribute substantive information. I suggest such a statement be posted in the discussion or in the statements section.
Introduction
paragraph 1 – L34 - Please add information about stress-related changes in the body (psychosomatic changes).
I suggest reading the publications and referring to them:
- stress linked to higher risk of cardiovascular disease - DOI: 10.1038/nrcardio.2012.45
- the effect of stress on the muscular system and the formation of muscular asymmetry - DOI: 10.3390/jcm10163459
- effects of stress on the digestive system - DOI: 10.1016/j.jpsychores.2021.110694
Materials and Methods
L74-75 – (ChronoSleep study reference number: –– ) - please provide consent numer.
L80-83 - Since the authors refer in Materials and Methods to the COVID-19 pandemic, he suggests adding a paragraph about stress changes during this period in the introduction.
For example:
- DOI: 10.3389/fpsyt.2022.828379
- DOI: 10.1016/j.jad.2020.08.001
- DOI: 10.1002/brb3.2318
L94 – ‘’[10.2196/preprints.46361]’’ - I suggest replacing the normal form of the quote
L107 – ‘’ Prism.’’ - Please provide exact data program, version and company.
Discussion
In the discussion, I would suggest adding more hypothetical considerations about the reactions of the nervous system, (physiological reactions) that may be related to the connection between stress and sleep.
Conflicts of Interest: - please choose your statement clearly.
Kind regards,
Piotr Gawda
Author Response
Thank you for your revision, which greatly improved the quality of the manuscript. The suggested changes are highlighted in the revised manuscript.
Abstract
L18 – ‘’We used a combination of Fitbit wearable actigraphy data and ‘’ - I suggest adding how much actigraphy was carried for.
Response: Unfortunately, the abstract only allows 200 words, therefore we can provide no details on the method
L22 -23 - The last sentence of the abstract suggests deleting. It does not contribute substantive information. I suggest such a statement be posted in the discussion or in the statements section.
Response: We changed this sentence.
Introduction
paragraph 1 – L34 - Please add information about stress-related changes in the body (psychosomatic changes).
I suggest reading the publications and referring to them:
- stress linked to higher risk of cardiovascular disease - DOI: 10.1038/nrcardio.2012.45
- the effect of stress on the muscular system and the formation of muscular asymmetry - DOI: 10.3390/jcm10163459
- effects of stress on the digestive system - DOI: 10.1016/j.jpsychores.2021.110694
Response: This has been added
Materials and Methods
L74-75 – (ChronoSleep study reference number: –– ) - please provide consent numer.
Response: The given number refers to the consent number form the ethics committee. We changed the term.
L80-83 - Since the authors refer in Materials and Methods to the COVID-19 pandemic, he suggests adding a paragraph about stress changes during this period in the introduction.
For example:
- DOI: 10.3389/fpsyt.2022.828379
- DOI: 10.1016/j.jad.2020.08.001
- DOI: 10.1002/brb3.2318
Response: We added these references to the discussion where we address the contribution of Covid-19 to psychosocial stress.
L94 – ‘’[10.2196/preprints.46361]’’ - I suggest replacing the normal form of the quote
Response: We replaced the form of the quote.
L107 – ‘’ Prism.’’ - Please provide exact data program, version and company.
Response: We added Version and company.
Discussion
In the discussion, I would suggest adding more hypothetical considerations about the reactions of the nervous system, (physiological reactions) that may be related to the connection between stress and sleep.
Response: We added this to the discussion.
Conflicts of Interest: - please choose your statement clearly.
Response: Done